# Precision-Controlled Bionic Lung Simulator for Dynamic Respiration Simulation

**DOI:** 10.3390/bioengineering12090963

**Published:** 2025-09-07

**Authors:** Rong-Heng Zhao, Shuai Ren, Yan Shi, Mao-Lin Cai, Tao Wang, Zu-Jin Luo

**Affiliations:** 1School of Automation, Beijing Institute of Technology, Beijing 100081, China; zhaorongheng@bit.edu.cn (R.-H.Z.); wangtaobit@bit.edu.cn (T.W.); 2School of Automation Science and Electrical Engineering, Beihang University, Beijing 100191, China; shiyan@buaa.edu.cn (Y.S.); caimaolin@buaa.edu.cn (M.-L.C.); 3Department of Respiratory and Critical Care Medicine, Beijing Engineering Research Center of Respiratory and Critical Care Medicine, Beijing Institute of Respiratory Medicine, Beijing Chao-Yang Hospital, Capital Medical University, Beijing 100029, China

**Keywords:** bionic lung simulator, ventilator, respiration simulation, lung compliance, airway resistance

## Abstract

Mechanical ventilation is indispensable for patients with severe respiratory conditions, and high-fidelity lung simulators play a pivotal role in ventilator testing, clinical training, and respiratory research. However, most existing simulators are passive, single-lung models with limited and discrete control over respiratory mechanics, which constrains their ability to reproduce realistic breathing dynamics. To overcome these limitations, this study presents a dual-chamber lung simulator that can operate in both active and passive modes. The system integrates a sliding mode controller enhanced by a linear extended state observer, enabling the accurate replication of complex respiratory patterns. In active mode, the simulator allows for the precise tuning of respiratory muscle force profiles, lung compliance, and airway resistance to generate physiologically accurate flow and pressure waveforms. Notably, it can effectively simulate pathological conditions such as acute respiratory distress syndrome (ARDS) and chronic obstructive pulmonary disease (COPD) by adjusting key parameters to mimic the characteristic respiratory mechanics of these disorders. Experimental results show that the absolute flow error remains within ±3 L/min, and the response time is under 200 ms, ensuring rapid and reliable performance. In passive mode, the simulator emulates ventilator-dependent conditions, providing continuous adjustability of lung compliance from 30 to 100 mL/${\mathrm{cmH}}_{2}$O and airway resistance from 2.01 to 14.67$\;\sqrt{{\mathrm{cmH}}_{2}O}/\left(L/s\right)$, with compliance deviations limited to ±5%. This design facilitates fine, continuous modulation of key respiratory parameters, making the system well-suited for evaluating ventilator performance, conducting human–machine interaction studies, and simulating pathological respiratory states.

## 1. Introduction

The global population’s accelerated aging and the high prevalence of pulmonary diseases have made ventilators indispensable in contemporary clinical practice. The precision and appropriateness of ventilator parameter settings directly influence the safety and efficacy of mechanical ventilation, imposing more stringent demands on ventilator design, manufacturing, and performance testing [1,2,3,4,5,6,7]. Lung simulators not only function as essential tools for evaluating ventilator performance but are also widely applied in clinical training and respiratory research [8,9,10,11,12,13]. Although animal experiments have historically played a crucial role in investigating respiratory physiology and verifying ventilator functionality, concerns related to ethics, costs, and reproducibility have increasingly driven the development of viable alternatives. Lung simulators offer a promising substitute by providing precisely controllable parameters such as tidal volume, airway pressure, and the respiratory rate. In contrast to animal models, which inherently vary due to differences in body weight, age, and health status, lung simulators facilitate standardized and repeatable testing across a wide range of respiratory conditions. The COVID-19 pandemic has further underscored such demands, driving extensive clinical trials and exploring AI-optimized combination therapies [14]. These advantages enhance experimental consistency, lower costs, and accelerate device development, aligning with current regulatory trends that advocate for minimizing or replacing animal testing in biomedical research [15,16,17].

Lung simulators are generally classified into two categories: passive and active systems. At present, passive lung simulators are more widely used in ventilator testing platforms due to their low cost and structural simplicity. These devices typically employ a splinted structure with fixed elasticity to simulate lung compliance and integrate ball valves at the air inlet to replicate airway resistance. They support ventilator performance assessment by enabling pressure and flow measurements at both the inlet and outlet. One prominent example is the SmartLung Adult lung simulator, developed by IMT Analytics Inc. However, despite their affordability and ease of construction, passive lung simulators rely heavily on the inherent physical characteristics of their constituent materials. As a result, they lack the ability to replicate patient-initiated spontaneous breathing and do not support continuous modulation of respiratory mechanics. Consequently, their use is limited in evaluating ventilator–patient synchrony performance under dynamic conditions [18,19,20].

In contrast, active lung simulators have been under development for nearly two decades, with the ASL 5000 by Ingmar Medical representing one of the most prominent examples. Dexter [21] validated the ASL 5000’s effectiveness in replicating neonatal and pediatric lung conditions, thereby underscoring its pivotal role in respiratory research. It has been utilized to assess the performance of innovative respiratory flow sensors by reproducing realistic breathing scenarios, thus providing a controlled environment to ensure sensor reliability under practical conditions [22]. Moreover, the ASL 5000 has been extensively applied in studies of assisted ventilation to simulate scenarios such as apnea and to examine the interaction between ventilators and lung models, facilitating the refinement of assisted ventilation techniques [23]. Consequently, the ASL 5000 makes a substantial contribution to the advancement of respiratory monitoring technologies and serves as a critical experimental platform for optimizing clinical ventilation strategies.

In addition to the ASL 5000, numerous researchers have explored various designs for active lung simulators. For example, Bunburaphong [24] developed an active lung simulator utilizing an airbag-spring mechanism and assessed its performance under Bi-Level Positive Airway Pressure (BiPAP) therapy. Elmaati [25] introduced a box–airbag configuration, derived transfer functions for three representative lung conditions—healthy lungs, acute respiratory distress syndrome (ARDS), and chronic obstructive pulmonary disease (COPD)—and conducted simulations using the developed simulator. Knöbel [26] implemented a cylinder–piston design to reproduce the respiratory process, assuming linear lung compliance, while Dong [27] from Nankai University employed a similar cylinder-piston architecture to construct a lung simulator for testing industrial oxygen masks. Notably, soft robotic devices for respiratory assistance, such as a diaphragm-assist device validated using a clinically relevant respiratory simulator, have also advanced respiratory care [28].

In addition, several researchers have explored hybrid active lung simulators that combine freshly excised porcine lungs with advanced inorganic materials to better replicate human respiratory physiology. While such approaches can improve biological fidelity, they are associated with considerable drawbacks, high operational costs and limited durability, as the excised pig lungs undergo irreversible degradation after only a few experimental cycles, necessitating frequent replacement [29,30,31,32].

Nevertheless, existing lung simulators continue to exhibit critical limitations. Specifically, (i) compliance and resistance are rarely adjustable in a continuous and precise manner, restricting their capacity to mimic diverse pathological states; (ii) dual active–passive operating modes are generally absent, limiting physiological versatility; and (iii) the ability to reproduce asymmetric dual-chamber dynamics and to evaluate patient–ventilator synchrony remains insufficient. These gaps highlight the urgent need for simulation systems that are both functionally adaptable and physiologically realistic.

In parallel, the growing demand for home care and portable ventilation has driven rapid progress in non-invasive ventilation devices and intelligent monitoring platforms. These developments not only broaden the clinical application landscape but also establish practical benchmarks for assessing lung simulators in terms of physiological fidelity and translational relevance. Although the present study is primarily oriented toward bench evaluation and research–teaching contexts, future work will aim to integrate the simulator with home care systems and portable ventilators, thereby expanding validation scenarios and enabling new directions in non-invasive ventilation design.

To address the aforementioned limitations, we propose a novel stepper motor-driven lung simulator. In this design, precise motor actuation directly controls airbag motion, effectively eliminating the hysteresis that is commonly observed in cylinder–piston mechanisms. By faithfully reproducing both active and passive respiratory behaviors, the simulator achieves enhanced accuracy, responsiveness, and configurability. These advantages make it a promising platform for ventilator performance testing and respiratory research, bridging the gap between existing simulation tools and the evolving demands of modern clinical practice. In future work, given the extensive dataset generated by the simulator, advanced deep learning time-series architectures such as LSTM–U-Net could be leveraged to classify ventilation modes and automatically detect anomalies, thereby enabling phase segmentation and the identification of patient–ventilator asynchrony patterns [33].

## 2. Methods

### 2.1. Concept

Lung compliance is a crucial indicator of respiratory system function and serves as a vital guide in clinical diagnosis and treatment. In the respiratory system, compliance $C_{L}\;\left(\mathrm{mL}/\mathrm{cm}H_{2}O\right)$ usually refers to the compliance of the lung tissue, and the formula can be tabulated as follows:(1)$$C_{L}=\frac{\Delta V}{\Delta p}$$

ΔV(mL) represents the change in lung volume. $\Delta p\;(\mathrm{cm}H_{2}O)$ represents the pressure difference in the lung.

Airway resistance is used clinically to reflect the degree of airway obstruction and assist in determining whether the cause of impaired pulmonary ventilation function is related to the airway [34]. The definition expression for airway resistance $R_{l}\;\left({\mathrm{cmH}}_{2}O/\left(L/s\right)\right)$ in medical terms is as follows:(2)$$R_{l}=\frac{\Delta p}{Q}$$

$\Delta p\;({\mathrm{cmH}}_{2}O)$ represents the pressure difference between the lung and the airway inlet. Q(L/s) represents airflow.

In clinical contexts, airflow within the human respiratory tract comprises both laminar and turbulent flow regimes. Turbulent flow typically arises in the larger airways (e.g., the trachea and main bronchi), where higher flow velocities and larger diameters are present, while laminar flow is predominant in the smaller peripheral airways. Classical definitions of airway resistance are generally derived under the assumption of laminar flow, in which gas moves in a smooth and orderly manner, resulting in resistance that is linearly proportional to flow velocity. However, this simplification fails to fully capture the complex airflow dynamics observed under physiological conditions.

In this study, an equivalent thin-walled orifice flow resistance model was employed, wherein the airway resistance was adjusted by modifying the orifice diameter. This configuration readily induces localized turbulent flow as air passes through the orifice, especially at elevated flow rates. According to principles of fluid mechanics, under turbulence-dominated conditions, the pressure drop across the resistive element is approximately proportional to the square of the flow velocity [35,36]. Therefore, to accurately characterize the resistive behavior of this component, an equivalent resistance coefficient $R\;(\sqrt{{\mathrm{cmH}}_{2}O}/\left(L/s\right))$ based on the orifice flow model is defined as follows:(3)$$R=\frac{\sqrt{\Delta p}}{Q}$$

$\Delta p\;({\mathrm{cmH}}_{2}O)$ represents the pressure difference across the two ends of the airway resistance adjustment component, and Q(L/s) denotes the flow rate through the airway resistance adjustment component.

### 2.2. System Structure

The lung simulator is shown in Figure 1. The stepper motor used in the experiment is the FSK40 model from Fuyu Technology Ltd. (Chengdu, China), and the stepper motor driver is the FMDD50D40NOM model from the same company. Additionally, the experiment employs NI cRIO-9053 from the National Instruments Corporation (Austin, TX, USA), as the controller, the FS6122 series flow-pressure sensors from Siargo Ltd. (Silicon Valley, CA, USA) for data acquisition, and the GII BiPAP ventilator from BMC Medical Co., Ltd. (Beijing, China) for testing the passive mode of the lung simulator.

In the proposed lung simulator system, the base of the airbag is fixed at the same level as the bottom of the linear guide rail, while the top of the airbag is attached to a stepper motor. The control parameters are input through the front panel interface of the host computer using LabVIEW. The controller receives these parameters and transmits corresponding drive signals to the stepper motor driver, which actuates vertical motion along the lead screw, thereby moving the airbag up and down. A flow-pressure sensor continuously collects the relevant flow and pressure signals, feeds them back to the controller, and transmits the data for real-time display on the LabVIEW interface. The overall system architecture is illustrated in Figure 2.

The airway resistance structure of the lung simulator designed in this study is illustrated in Figure 3.

Beyond the immediate focus on respiratory dynamics and control, the long-term durability of the airbag and resistance components remains a critical consideration. In the present study, fuzzy logic–based fatigue assessment could not be implemented due to the lack of long-duration stress–damage datasets. As an engineering substitute, periodic cyclic fatigue testing combined with baseline re-measurement of compliance and the leakage rate is planned to monitor mechanical degradation. Looking ahead, with sufficient experimental data, intuitionistic fuzzy divergence methods may be applied to capture uncertain stress states and improve elastomer fatigue prediction, thereby enhancing the reliability of the simulator in prolonged operation.

The airway resistance is adjusted by controlling the extension of the motor, which adjusts the orifice. The airway resistance adjustment component of the SmartLung Adult lung simulator is used as a reference. The resistance adjustment component of the lung simulator was tested using a mass flow controller, with the setup shown in Figure 4. Gas flows from the compressed air source through the mass flow controller, then passes through the resistance adjustment component of the lung simulator, and finally exits into the atmosphere.

When the flow rate was set to 0.33, 0.50, 0.66, and 0.83 L/s, the pressure differential across the resistance adjustment component of the SmartLung Adult lung simulator was measured. The corresponding resistance coefficients were then calculated, as presented in Table 1 and Table 2. As shown in the results, for each resistance setting, the calculated resistance coefficient r remained relatively stable across different flow rates, indicating that r does not vary significantly with changes in airflow. These suggest that r is primarily determined by the opening level of the resistance adjustment component rather than the flow rate itself, thereby providing a reliable representation of the mechanical resistance setting of the simulator. These findings support the validity and rationality of the defined resistance coefficient r as a descriptor of the lung simulator’s adjustable resistance characteristics.

Using an air compressor as the gas source, pass a constant flow rate of 30 L/min through the gas resistance adjustment device via a mass flow controller. Adjust the size of the throttle opening, measure the pressure difference on both sides of the device, and calculate the gas resistance. The resistances *R* corresponding to different throttle opening radii *r* (mm) are shown in Figure 5. The fitting equation is as follows:(4)$$R=10.82\times r^{-1.01}$$

### 2.3. Control Methods

To provide a clear framework for the proposed control strategies, this section is structured into two complementary parts. Section 2.3.1 introduces the passive mode, in which the simulator’s compliance is precisely regulated through motor-driven airbag dynamics under an observer-based sliding mode control scheme. Section 2.3.2 then presents the active mode, which establishes a lumped-parameter physiological model of the dual-chamber lungs to reproduce realistic flow–pressure interactions. Together, these two modes form the foundation of our innovation by integrating compliance-targeted precision with physiologically representative dynamics, thereby ensuring both experimental reproducibility and clinical relevance.

#### 2.3.1. Passive Mode

When the lung simulator is in passive mode, the airbag functions as an expandable cylinder, with the stored air volume inside the airbag representing its volume. Neglecting the gas compression caused by pressure variations inside the airbag, the airflow rate into the safety airbag can be expressed by the following equation:(5)$$Q=vs_{d}/10^{3}$$
$s_{d}$ ($cm^{2}$) represents the base area of the airbag, v(cm/s) represents the operating velocity of the motor. Due to the inherent compliance of the airbag, lateral deformation occurs when the internal pressure of the airbag changes. Define the internal disturbance d of the system. Therefore, (Equation 5) can be written as follows:(6)$$Q=\left(vs_{d}+d\right)/10^{3}$$

Thus, we can obtain the tidal volume $V_{t}$(mL) generated by breathing:(7)$$V_{t}=\int \left(Q\cdot 10^{3}\right)$$

The total air volume inside the airbag is $V=V_{t}+V_{F}$, where $V_{F}$ (mL) is the residual volume. In this study, a linear compliance model is employed, where the rate of lung volume change concerning intrapulmonary pressure remains constant over time. Consequently, compliance can be equivalently represented as the ratio of lung volume to intrapulmonary pressure. We can obtain the device compliance $C\;(\mathrm{mL}/\mathrm{cm}H_{2}O)$ of the lung simulator as follows:(8)$$C=\frac{V}{p}$$
$p\;(\mathrm{cm}H_{2}O)$ represents the pressure inside the airbag. Let x=V, u=v, $b=s_{d}$, c=1/p, and y=C; the system input is the motor velocity, and the output is the system compliance. According to the design objectives, the system must be maintained at the target compliance $C_{d}\;\left(\mathrm{mL}/\mathrm{cm}H_{2}O\right)$, $y_{d}=C_{d}$, which means the system must be stabilized at the target state xd. The state equation for the lung simulator system is given by the following equation:(9)$$\begin{array}{c} \dot{x}=bu+d\\ y=cx \end{array}$$

By extending the system shown in (Equation 9), we obtain the following:(10)$$\begin{array}{c} {\dot{x}}_{1}=x_{2}+bu\\ {\dot{x}}_{2}=h \end{array}$$
$x_{2}=d$ is the disturbance, and *h* is the derivative of the disturbance. Design a linear extended state observer (LESO) for the system described by the following equations (Equation 11):(11)$$\begin{array}{c} e_{1}=x_{1}-z_{1}\\ e_{2}=x_{2}-z_{2}\\ {\dot{z}}_{1}=z_{2}+bu+\beta_{1}\mathrm{fal}(e_{1},\alpha_{1},\delta )\\ {\dot{z}}_{2}=\beta_{2}\mathrm{fal}(e_{1},\alpha_{2},\delta )\\ {\dot{e}}_{1}=e_{2}-\beta_{1}\mathrm{fal}\left(e_{1},\alpha_{1},\delta \right)\\ {\dot{e}}_{2}=h-\beta_{2}\mathrm{fal}\left(e_{1},\alpha_{2},\delta \right) \end{array}$$
$z_{i}\left(i=1,2\right)$ is the observed value of $x_{i}\left(i=1,2\right)$, and $\beta_{i}\left(i=1,2\right)$ is the observation gain.(12)$$\mathrm{fal}\left(e,\alpha ,\delta \right)=\left\{\begin{array}{c} {\left|e\right|}^{\alpha }\mathrm{sign}\left(e\right),\left|e\right|>\alpha \\ e/\delta^{1-\alpha },\left|e\leq \delta \right| \end{array}\right.$$(13)$$\mathrm{sign}\left(e\right)=\left\{\begin{array}{c} 1,e>0\\ 0,e=0\\ -1,e<0 \end{array}\right.$$

Design the sliding surface for the observed state of the linear extended state observer, with the error between the observed state and the target state as follows:(14)$$e=z_{1}-x_{d}$$

The design of the sliding surface is as follows:(15)s=e+k∫e
k>0 is the sliding mode area integral coefficient. The design of the sliding mode surface-reaching law is as follows:(16)$$\begin{array}{c} \dot{s}=-k_{1}s-sigmoid\left(s\right){\left|s\right|}^{0.5}\\ sigmoid\left(s\right)=1/\left(1+e^{-s}\right) \end{array}$$

The compliance control block diagram of the lung simulator system is shown in Figure 6.

Traditional SMC typically utilizes the discontinuous sign function, sign(s) resulting in high switching frequency and significant chattering. A smooth S-shaped function, the sigmoid function, is introduced to address this issue. The sigmoid function is continuously differentiable and exhibits the following asymptotic behavior: $\lim_{s\to +\infty }\mathrm{sigmoid}\left(s\right)=1,$$\lim_{s\to -\infty }\mathrm{sigmoid}\left(s\right)=0$. When s approaches zero, sigmoid(s) varies smoothly, substantially reducing chattering. Furthermore, in the presence of significant tracking errors, it maintains sufficient control effort to drive the system toward the sliding surface. This property ensures a desirable balance between convergence speed and steady-state accuracy. Therefore, substituting the traditional discontinuous sign(s) function with the smooth sigmoid(s) function can effectively suppress chattering and enhance the overall dynamic performance of the control system. $k_{1}>0$ is the proportional coefficient of the reaching law. Therefore, the SMC law is as follows:(17)$$\begin{array}{c} u=\frac{1}{b}(-k_{1}s-sigmoid\left(s\right){\left|s\right|}^{0.5}\\ -z_{2}-\beta_{01}\mathrm{fal}(e_{1},\alpha_{1},\delta )-ce) \end{array}$$

For the sliding mode controller, select the Lyapunov function $\dot{V}=\frac{1}{2}s^{2}+\frac{1}{2}e_{1}^{2}+\frac{1}{2}e_{2}^{2}$. Thus, we obtain the following:(18)$$\begin{array}{c} \dot{V}=s\dot{s}+e_{1}{\dot{e}}_{1}+e_{2}{\dot{e}}_{2}\\ =s(-k_{1}s-sigmoid\left(s\right){\left|s\right|}^{0.5})\\ +e_{1}\left(e_{2}-\beta_{1}\mathrm{fal}\left(e_{1},\alpha_{1},\delta \right)\right)\\ +e_{2}\left(h-\beta_{2}\mathrm{fal}\left(e_{1},\alpha_{2},\delta \right)\right) \end{array}$$

For ∀s, $\beta_{1}>\frac{e_{2}}{\mathrm{fal}(e_{1},\alpha_{1},\delta )},\beta_{2}>\frac{h}{\mathrm{fal}(e_{1},\alpha_{2},\delta )}$, V≥0, we have $\dot{V}\leq 0$. According to the Lyapunov stability criterion, it can be inferred that the system is stable with the sliding mode control law shown in (Equation 17). The *fal* function is a nonlinear function characterized by fast convergence, exhibiting the properties of having a ‘*large error with a small gain, small error with a large gain*.’ Its inclusion can reduce response time and improve dynamic performance. Taking α=0.5 and δ=0.01 as examples, the fal function is shown in Figure 7.

The parameter values that are relevant to this section are provided in Table 3.

Taking $C_{d}=100\;\mathrm{mL}/\mathrm{cm}H_{2}O$ and $PEEP=4\;\mathrm{cm}H_{2}O$ as an illustrative example, with an initial condition *s*(0) = 400, the time-domain response of *s*(t) is depicted in the Figure 8 below. The convergence time of the sliding surface is approximately $t_{SMC}\approx 300\;\mathrm{ms}$.

The convergence time of the observer is determined by $t_{LESO}\approx 1/\beta_{1}$, and according to the parameters listed in Table 1, $t_{LESO}\approx 10\mathrm{ms}$.

Given that $t_{LESO}<<t_{SMC}$, the separation principle is satisfied, thereby allowing the independent design of the observer and the controller without compromising the overall stability and performance of the closed-loop system.

The actual control process is discrete. The control cycle is set to 1 ms to meet the control accuracy requirements. Since 1 ms is markedly shorter than $t_{SMC}$, it does not lead to chattering phenomena or destabilize the system. Discretizing the equations mentioned above, the critical formulas involved are as follows:(19)$$\begin{array}{c} z_{1}\left(k+1\right)=z1\left(k\right)+h_{1}\left(z_{2}\left(k\right)+\beta_{01}\mathrm{fal}\left(e\left(k\right),\alpha_{1},\delta \right)+bu\left(k\right)\right) \end{array}$$(20)$$z_{2}\left(k+1\right)=z_{2}\left(k\right)+h_{1}\beta_{02}\mathrm{fal}\left(e\left(k\right),\alpha_{2},\delta \right)$$
$h_{1}$ represents the control cycle.

#### 2.3.2. Active Mode

To simulate human respiratory function, it is necessary first to model and simulate the physiological system of the lungs. The respiratory system consists of the nose, pharynx, larynx, trachea, bronchi, and lungs. This study focuses primarily on the mechanical simulation of the pulmonary physiological environment; therefore, the nasal, pharyngeal, and laryngeal components are not considered.

Based on the above analysis, a lumped-parameter model is constructed, as shown in Figure 9. $R_{t}\;\left(\mathrm{cm}H_{2}O/L/s\right)$ represents the upper airway resistance, $R_{l}\left(\mathrm{cm}H_{2}O/L/s\right)$ and $R_{r}\;\left(\mathrm{cm}H_{2}O/L/s\right)$ represent the small airway resistances of the left and right lungs, respectively, $C_{ll}\;\left(\mathrm{mL}/\mathrm{cm}H_{2}O\right)$, $C_{lr}\;\left(\mathrm{mL}/\mathrm{cm}H_{2}O\right)$, and $C_{cw}\;\left(\mathrm{mL}/\mathrm{cm}H_{2}O\right)$ represent the compliance of the left lung, right lung, and chest wall, respectively. In contrast, $P_{l}\;\left(\mathrm{cm}H_{2}O\right)$, $P_{pl}\;\left(\mathrm{cm}H_{2}O\right)$, and $P_{mus}\;\left(\mathrm{cm}H_{2}O\right)$ mean the intrapulmonary pressure, chest wall pressure, and respiratory muscle force, respectively. Q(L/min), $Q_{l}\;\left(L/\min\right)$, and $Q_{r}\;\left(L/\min\right)$ represent the airflow in the upper airway and the left and right lungs, respectively. All the aforementioned parameters can be freely adjusted in this model.

In the present study, the musculoskeletal contribution of the chest wall is represented through a lumped-parameter element $C_{cw}$, which was selected to ensure interpretability and maintain the real-time control performance of the simulator. This simplified treatment is consistent with the primary objective of reproducing respiratory dynamics in a controllable and computationally efficient manner. Nevertheless, we acknowledge that finite element modelling (FEM) offers clear advantages in capturing detailed musculoskeletal interactions and complex biomechanical deformations of the chest wall. Although direct FEM integration is less compatible with the current device architecture due to its computational complexity and reduced responsiveness, FEM-based modules are envisioned in future development as optional high-fidelity extensions. These could be interfaced with the lumped-parameter model through boundary conditions and parameter exchange, thereby enabling posture-specific and pathology-specific simulations. Such a layered strategy preserves present usability while providing a pathway toward enhanced biomechanical realism.

In lung function testing, the focus is often on the upper airway flow. Similarly, in the lung simulator designed in this study, the primary focus is on the upper airway flow and intrapulmonary pressure. Based on the electrical model shown in Figure 9, the input–output transfer function (Equation 21) is derived, where the input is the respiratory muscle force, and the output is the upper airway flow.(21)$$\begin{array}{c} \frac{Q(s)}{P_{mus}\left(s\right)}=-60\cdot \frac{g_{4}s^{2}+g_{5}s}{g_{1}s^{2}+g_{2}s+g_{3}}\\ f_{1}=R_{l}\cdot R_{r}\cdot C_{ll}\cdot C_{lr}\\ f_{2}=R_{l}\cdot C_{ll}+R_{r}\cdot C_{lr}\\ f_{3}=R_{l}\cdot C_{ll}\cdot C_{lr}+R_{r}\cdot C_{ll}\cdot C_{lr}\\ f_{4}=C_{ll}+C_{lr}\\ g_{1}=f_{1}\cdot C_{cw}+f_{3}\cdot C_{cw}\cdot R_{t}\\ g_{2}=f_{3}+f_{2}\cdot C_{cw}+f_{4}\cdot C_{cw}\cdot R_{t}\\ g_{3}=f_{4}+C_{cw}\\ g_{4}=f_{3}\cdot C_{cw}\\ g_{5}=f_{4}\cdot C_{cw} \end{array}$$

Equation (Equation 21) represents the frequency-domain transfer function. In practical control processes, converting this into the time domain and determining the time-domain response with an initial value of zero is necessary.

The lung simulator device’s target flow output is the upper airway flow. An open-loop control approach is employed to ensure the system’s stability, where the flow rate is controlled by adjusting the operating speed of the electric cylinder. The position of the electric cylinder and tidal volume are calibrated. The calibration results are shown in (Equation 22).(22)$$\begin{array}{c} x=-2.9205V_{t}^{3}+4.6297V_{t}^{2}+5.6886V_{t}+0.423 \end{array}$$

The relationship between the electric cylinder’s operating speed and the target flow can be derived as (Equation 23).(23)$$\begin{array}{c} \begin{array}{c} k=\frac{x}{V_{t}}\\ v=\frac{k\cdot Q}{60} \end{array} \end{array}$$

It is important to note that in the electrical analog model, the resistance of the upper airway is represented by a linear element. However, in the actual resistance regulation device (as shown in Figure 3), the flow resistance exhibits nonlinear characteristics due to turbulence. To account for this discrepancy in practical computations, a custom-defined resistance coefficient r is introduced to replace the theoretical resistance Rt equivalently. Accordingly, the calculation of the target intrapulmonary pressure is modified as follows:(24)$$\begin{array}{c} P_{l}=Q^{2}\cdot r^{2} \end{array}$$

## 3. Results

### 3.1. Simulation

A R2025a MATLAB/Simulink-based lung simulator simulation platform was developed to verify the effectiveness of the proposed compliance and airway resistance control algorithm. The main ventilator parameter settings are listed in Table 4. $P_{aw}$ represents the pressure generated by the ventilator. The simulation uses the following model:(25)$$\begin{array}{c} P_{aw}=V/C+r^{2}{\dot{V}}_{t}^{2} \end{array}$$

The simulation model was independently constructed based on (Equation 25) rather than employing existing Simscape medical ventilator templates. Specifically, the compliance control section was implemented using the discretized formulations (Equation 19) and (Equation 20), with relevant parameters set according to Table 3, while the airway resistance control section was realized following (Equation 3). This implementation ensures consistency with the theoretical framework described earlier and allows for direct validation against the proposed hardware platform.

Figure 10 compares the simulation results and the experimental data obtained from the physical platform. All data were recorded after the system reached a steady-state condition. Under the prescribed lung compliance and airway resistance settings, the difference in compliance values between the simulation and experiment remained within ±5%, indicating good agreement and validating the model’s ability to reproduce pulmonary mechanical behavior accurately. However, when calculating airway resistance using (Equation 3), significant deviations were observed when the flow rate *Q* approached zero or remained at very low levels. This discrepancy is primarily attributed to the mathematical structure of the equation, where the presence of flow in the denominator causes even minimal measurement noise in pressure or flow to be amplified under low-flow conditions. As a result, the computed resistance becomes unstable and deviates from its actual physical meaning. Moreover, the flow characteristics within the thin-walled orifice structure are susceptible to changes in flow velocity. At low flow rates, the airflow typically falls within the transitional region between laminar and turbulent regimes, where the instability of the flow pattern directly affects the pressure–flow relationship. These introduce nonlinear disturbances and further increase the uncertainty in resistance estimation.

### 3.2. Passive Mode

In passive mode, the lung simulator device mimics the breathing pattern of a human body that relies on a ventilator when the person is unconscious. The ventilator parameters are set as follows: $P_{peak}$ = 15 ${\mathrm{cmH}}_{2}$O, PEEP = 4 ${\mathrm{cmH}}_{2}$O, RR = 15, $V_{F}$ = PEEP × $C_{d}$.

During ventilation, the lung simulator is connected to the ventilator via a three-way connector. The test results in passive mode are presented in Figure 11, Figure 12, Figure 13 and Figure 14. It is important to note that, according to the classical theoretical Formulation (Equation 1), compliance is expected to approach zero when the tidal volume is zero. However, the compliance calculation method adopted in this study is based on (Equation 8), which defines compliance as the ratio of lung volume to intrapulmonary pressure. Under this definition, the computed compliance does not equal zero even when the tidal volume is zero.

Over three cycles, at the beginning of inhalation and exhalation, there is significant jitter; the measurement values, compared to the target compliance, are presented in Table 5. Calculations show that the errors are within ±5%.

### 3.3. Active Mode

The model simulation data and the measured data from the lung simulator output are shown in Figure 15. The respiratory measurement data under different respiratory muscle force conditions are shown in Figure 16. In the experiment, lung parameters under different severities of ARDS and COPD were preset. The respiratory measurement data are shown in Figure 17 and Figure 18. The parameter settings are detailed in Table 6, Table 7, Table 8 and Table 9.

In Figure 15, $q_{V-s}$, $P_{l-s}$, and $V_{t-s}$ represent the simulated flow, intrapulmonary pressure, and tidal volume values, respectively. In contrast, $q_{V-m}$, $P_{l-m}$, and $V_{t-m}$ represent the measured flow, intrapulmonary pressure, and tidal volume values, respectively. There is a delay between the simulated and measured values, known as the response time. Under different parameter settings, the response times (ms) are shown in Table 10, with the overall response time maintained within 200 ms.

The primary controlled variable in the lung simulator device is the flow rate. The absolute error (L/min) between the measured flow rate and the target flow rate under different parameter settings is shown in Table 10. The absolute overall mistake is maintained within 3 L/min.

## 4. Discussion

In passive mode, during ventilation, the ventilator connects to the left and right lungs via a three-way connector. Due to the elasticity of the tubing, even under conditions that strive to keep the tubing as straight as possible, airflow through the three-way connector can still experience some pressure imbalance, leading to different resistances on each side. These are illustrated in Figure 11. The figure shows the respiratory parameter data for the left and right lungs under identical compliance and resistance settings. Although the parameters are similar, the respiratory parameter data for the two lungs differ. Specifically, the tidal volume on the right side is significantly larger than on the left side, indicating that the resistance on the left side is more significant than on the right. Typically, an increase in resistance would lead to a prolonged expiratory phase, as shown in Figure 14. However, an increase in resistance shortens the expiratory phase. This phenomenon is caused by the effect illustrated in Figure 19.

Figure 19 shows the isobar distribution simulated using COMSOL software (6.2.0.658 Version) under conditions of unequal resistance on both sides of a three-way connector. Airflow enters at a certain speed from the inlet, and at the two outlet ducts, the pressure distribution results in $p_{1}>p_{2}$ and $p_{3}<p_{4}$ due to the unequal resistance. From $p_{3}<p_{4}$, it can be concluded that the pressure inside the lung on the side with higher resistance is lower, which aligns with the actual situation. When exhalation begins, the airflow exits the lung, and the pressure drop is more significant on the side with higher resistance, which also has a smaller tidal volume. These conditions ultimately lead to a shortened expiratory phase on the side with higher resistance.

The ventilator supplied initial pressure in experiments involving different compliance for the left and right lungs, and PEEP (positive end-expiratory pressure) was set to 4 cmH_2_O. At this point, the pressure in both lungs could reach 4 cmH_2_O. When the ventilator began ventilating the lung simulator, the difference in compliance between the two lungs caused the lung with lower compliance to get the PEEP first after the first respiratory cycle. In contrast, the pressure in the other lung continued to decrease. This led to the jetting phenomenon shown in Figure 20.

Figure 20 illustrates the velocity distribution when airflow enters the three-way connector from point B at a constant speed. Although there is no airflow entering the symmetrical side at this time, it is evident that there are fluctuations in airflow velocity at point A, which will cause pressure changes on the opposite side. The jetting phenomenon caused the gas in the lung with lower compliance to continue flowing out, with the pressure dropping further below the PEEP set by the ventilator, ultimately resulting in a negative tidal volume in the lung with lower compliance. These conditions are also why the negative tidal volumes are observed in Figure 12. It is important to note that the negative tidal volume here can be interpreted as a decrease in residual gas in the lung with lower compliance.

Under conditions of constant airway resistance, increasing lung compliance leads to substantial changes in respiratory mechanics. As compliance rises, tidal volume increases accordingly, since the lungs can accommodate a greater volume of air under the same pressure, as described by (Equation 1). This enhanced elasticity reduces the mechanical load during inspiration, lowering the peak inspiratory pressure and decreasing overall respiratory effort. Furthermore, the resulting increase in tidal volume prolongs the pressure decay phase, which subsequently alters the airflow profile during the breathing cycle. These effects are depicted in Figure 13, where the temporal evolution of key respiratory parameters under varying compliance is presented.

Maintaining constant compliance while progressively increasing airway resistance elicits distinct dynamic respiratory responses. As described in (Equation 2), elevated airway resistance reduces tidal volume, since the increased resistance impedes airflow, thereby limiting the volume of gas delivered to the lungs under fixed pressure support. In the context of conscious, spontaneous breathing, greater airway resistance necessitates higher inspiratory pressure to achieve sufficient lung inflation, leading to increased peak airway pressures. However, in the device’s passive mode—which simulates unconscious, ventilator-dependent breathing with fixed pressure and inspiratory duration—rising airway resistance paradoxically results in a lower peak pressure, as less volume is delivered within the same time frame. During expiration, airflow decelerates and the overall breathing cycle is prolonged. These mechanical effects are further complicated by computational artifacts: when a parabolic resistance model is employed, the flow-dependent nature of resistance leads to significant numerical instabilities near zero-flow regions, especially during transitions between inspiration and expiration. These errors primarily arise from numerical discontinuities and reduced sensor resolution at low flow rates, ultimately undermining the accuracy of resistance estimation. Figure 14 illustrates these phenomena, highlighting both the physiological and numerical consequences of increasing airway resistance.

Figure 15a illustrates the respiratory muscle force curve, while Figure 15b–d depict the comparison between simulation data (black dashed lines) and measured data (red solid lines). A noticeable lag is observed between the simulation and measured data, primarily attributable to the system architecture design of the lung simulator device. The simulation data curve is initially derived from predefined parameters during operation. Subsequently, motor movement is controlled according to the simulation curve, and flow and pressure signals are collected by sensors. These signals are then processed by the controller and displayed on the PC. Figure 15b,d present the flow and tidal volume information, respectively, demonstrating that the measured and simulation data exhibit similar trends with closely matched amplitudes. Figure 15c presents the pressure profiles, where the overall trends in the experimental and simulation data are generally consistent; however, a noticeable discrepancy in amplitude is observed. This difference primarily arises from structural distinctions between the simulation model and the physical system. Furthermore, as illustrated in Figure 15b, the expiratory phase exhibits a rapid change in flow rate. The physical structure of the lung simulator is limited in its ability to accurately replicate this transient behavior, resulting in a higher peak expiratory flow in the simulation compared to that observed in the experimental data.

An increase in the maximum respiratory driving force results in elevated inspiratory and expiratory flow peaks, greater tidal volume, and larger absolute intrapulmonary pressure values. Notably, the expiratory peak consistently exceeds the inspiratory peak, a pattern that is attributable to the asymmetric time profile of the driving force, which rises gradually but decays rapidly. During the inspiratory phase, lung expansion is passive and constrained by elastic recoil and airway resistance, thereby limiting inspiratory flow. In contrast, expiration is predominantly driven by the swift release of stored elastic energy, with minimal muscular contribution. These dynamics are clearly depicted in Figure 16.

Under identical respiratory driving force conditions, the progression of acute respiratory distress syndrome (ARDS) alters respiratory behavior substantially. As ARDS severity increases, lung compliance decreases due to widespread alveolar collapse, while resistance increases, reflecting a reduction in the functional residual capacity (FRC). These pathological changes lead to diminished tidal volume, reduced airflow, a shortened exhalation time, and lower peak positive and negative intrapulmonary pressures during expiration, as evidenced by the data shown in Figure 17.

A similar trend is observed in chronic obstructive pulmonary disease (COPD), where escalating disease severity leads to increased upper airway and lung resistance alongside diminished elastic recoil. As a result, respiratory flow is reduced, exhalation is prolonged, and tidal volume decreases under the same respiratory muscle force conditions, as illustrated in Figure 18b,d. Furthermore, while the absolute magnitude of intrapulmonary pressure rises during both inspiration and expiration with increasing COPD severity, a divergence emerges: Figure 18c shows that inspiratory negative pressure increases, whereas expiratory positive pressure decreases. This counterintuitive outcome stems from the interaction between resistance and flow, as described by (Equation 23); when flow decreases more than resistance increases, the resulting intrapulmonary pressure can decline despite elevated resistance.

The experiment verified that the designed lung simulation system exhibits effective compliance control. Compared to traditional passive lung simulators, it offers a more comprehensive adjustable range of compliance and meets the requirements for continuous regulation.

While the present study demonstrates that the proposed lung simulator achieves effective compliance control and stable device performance, several directions remain open for future research. To enhance the spatial fidelity of airflow and pressure distribution, we plan to extend CFD-based cross-checks under measured boundary conditions and leverage reserved hardware/timing interfaces for the integration of multi-point pressure sensors or electrical impedance tomography (EIT). For long-term structural reliability, since the airbag and resistance components are primarily polymer-based elastomers and conventional eddy current testing is less suitable, future designs involving conductive composites may incorporate eddy current or ultrasonic NDT techniques. To facilitate downstream applications, we will establish a standardized data/label interface and logging protocol, enabling the simulator’s output to be applied to advanced methods such as LSTM- or U-Net–based breathing phase segmentation and patient–ventilator asynchrony detection. Furthermore, the integration of electronics and intelligent monitoring represents another promising avenue. The current system already supports remote telemetry and structured logging, allowing for threshold- and SPC-based prototype alerts for abnormal events. With the accumulation of long-term operational data, predictive maintenance strategies based on drift detection and anomaly distribution modeling can be developed, thereby providing early warnings of performance degradation and complementing the physiological and mechanical extensions described above.

## 5. Conclusions

This work presents a stepper motor-driven, active–passive integrated lung simulator that replicates human respiratory mechanics with high fidelity, employing a dual-chamber, lumped-parameter model to emulate asymmetric lung behavior dynamically. In passive mode, it enables continuous compliance adjustment within 30–100 mL/${\mathrm{cmH}}_{2}$O—surpassing conventional simulators’ static settings—with minor operational compliance fluctuations that are addressable via displacement sensors and advanced control algorithms. In active mode, driven by customizable respiratory muscle force curves, it supports autonomous breathing (up to 20 ${\mathrm{cmH}}_{2}$O of driving pressure) and accurately simulates ARDS (via reduced compliance and increased resistance for restrictive defects) and COPD (via elevated resistance and air trapping for obstructive patterns). It allows for the dynamic tuning of chest wall compliance (0–250 mL/${\mathrm{cmH}}_{2}$O), single-lung compliance (0–100 mL/${\mathrm{cmH}}_{2}$O), upper airway resistance (2.01–14.67 $\;\sqrt{{\mathrm{cmH}}_{2}O}/\left(L/s\right)$), and small airway resistance (2–200 ${\mathrm{cmH}}_{2}$O/(L/s)), with a control delay of <200 ms and a flow tracking error within ±3 L/min. Seamlessly transitioning between modes, this versatile platform aids respiratory pathophysiology research, ventilator optimization, and medical education, with ongoing work focusing on dual-chamber physical realization to enhance fidelity.

## Figures and Tables

**Figure 1 bioengineering-12-00963-f001:**
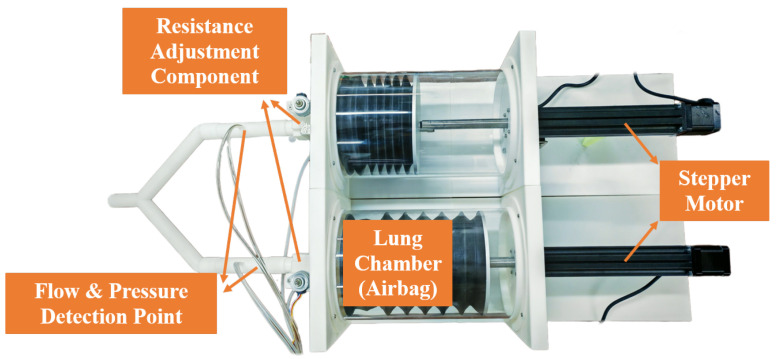
Lung Simulator Device.

**Figure 2 bioengineering-12-00963-f002:**
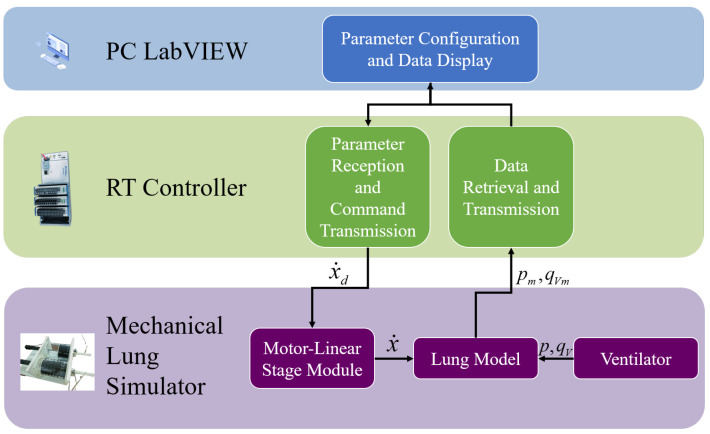
Overall System Layout Diagram. (${\dot{x}}_{d}$—demand velocity, $\dot{x}$—measured velocity, *p*—ventilator applied pressure, $q_{V}$—ventilator applied flow, $p_{m}$—measured pressure, $q_{Vm}$—measured flow).

**Figure 3 bioengineering-12-00963-f003:**
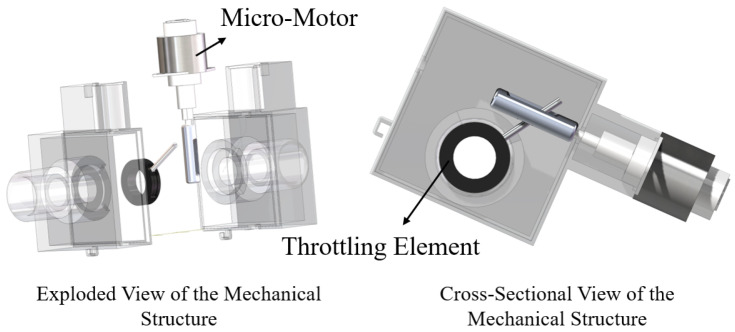
Resistance adjustment device.

**Figure 4 bioengineering-12-00963-f004:**
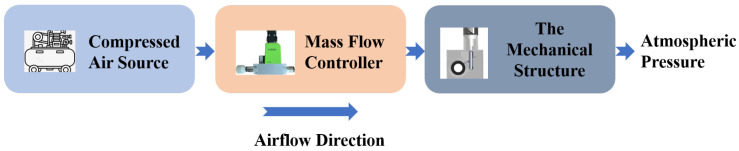
Airway resistance calibration system.

**Figure 5 bioengineering-12-00963-f005:**
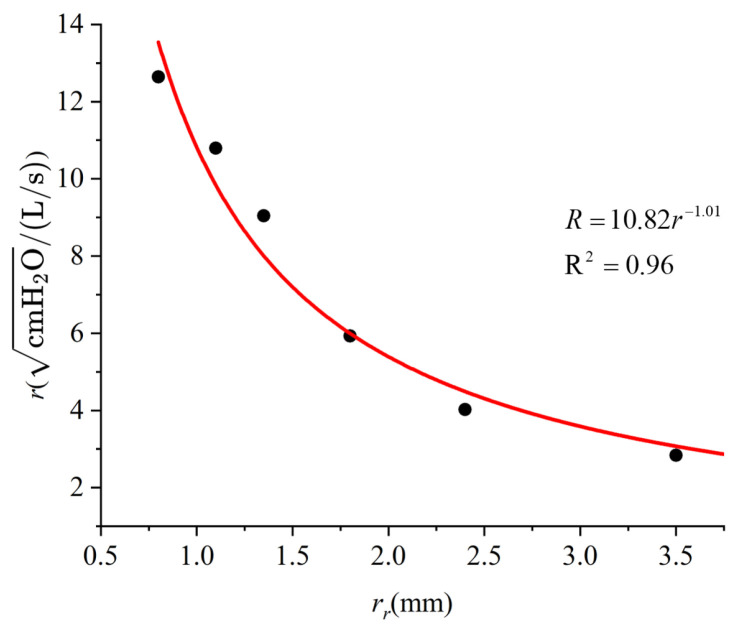
Relationship Between Throttle Radius and Airway Resistance.

**Figure 6 bioengineering-12-00963-f006:**
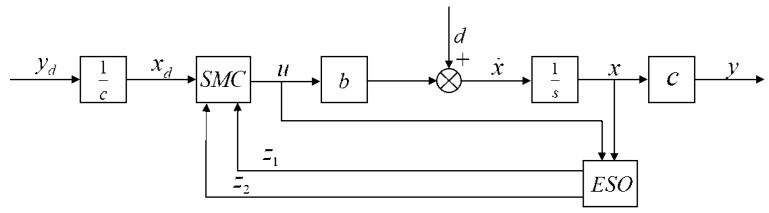
Structure diagram of sliding mode control based on extended state observer.

**Figure 7 bioengineering-12-00963-f007:**
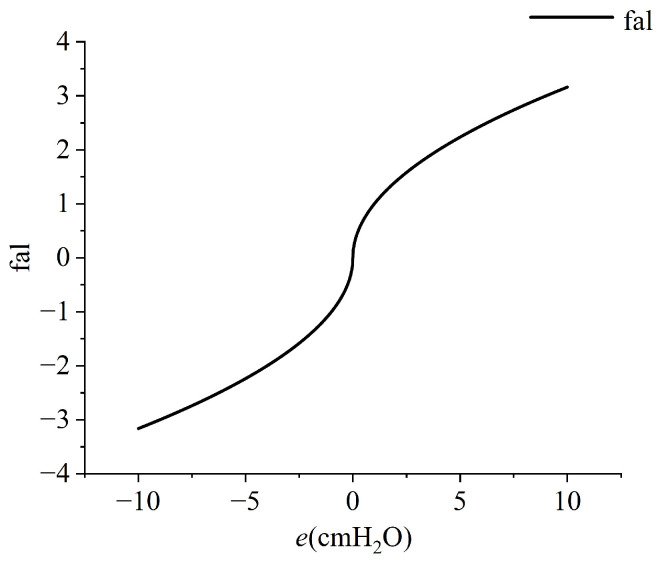
*fal* function.

**Figure 8 bioengineering-12-00963-f008:**
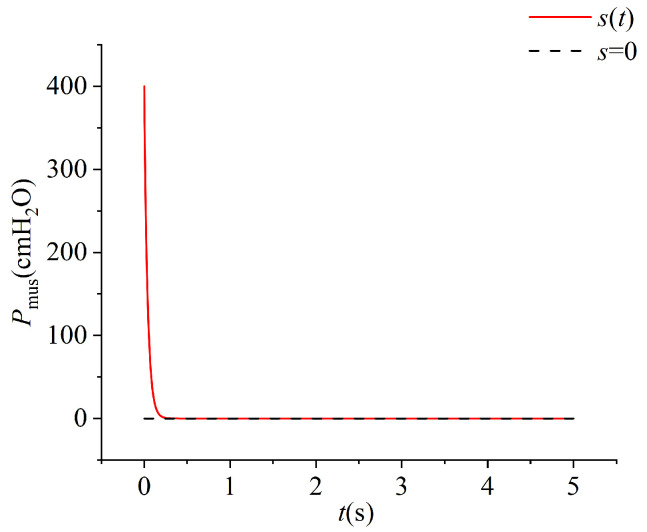
Time-domain response of the signal *s*(t).

**Figure 9 bioengineering-12-00963-f009:**
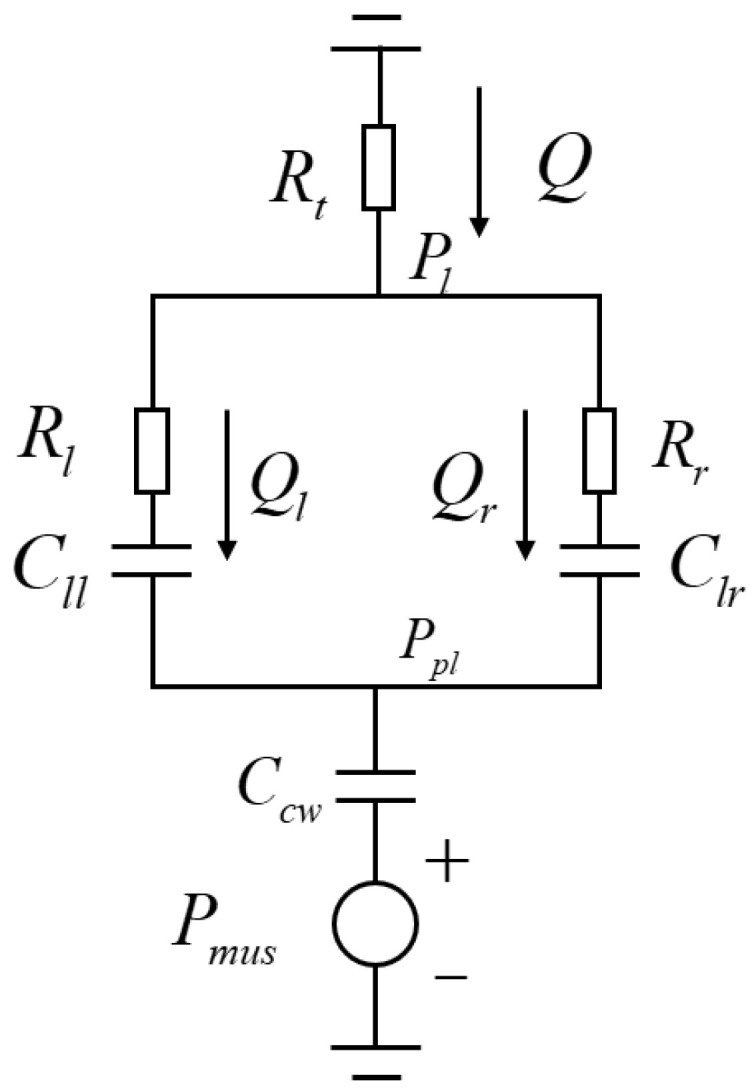
Dual-Chamber Lung Lumped Parameter Model.

**Figure 10 bioengineering-12-00963-f010:**
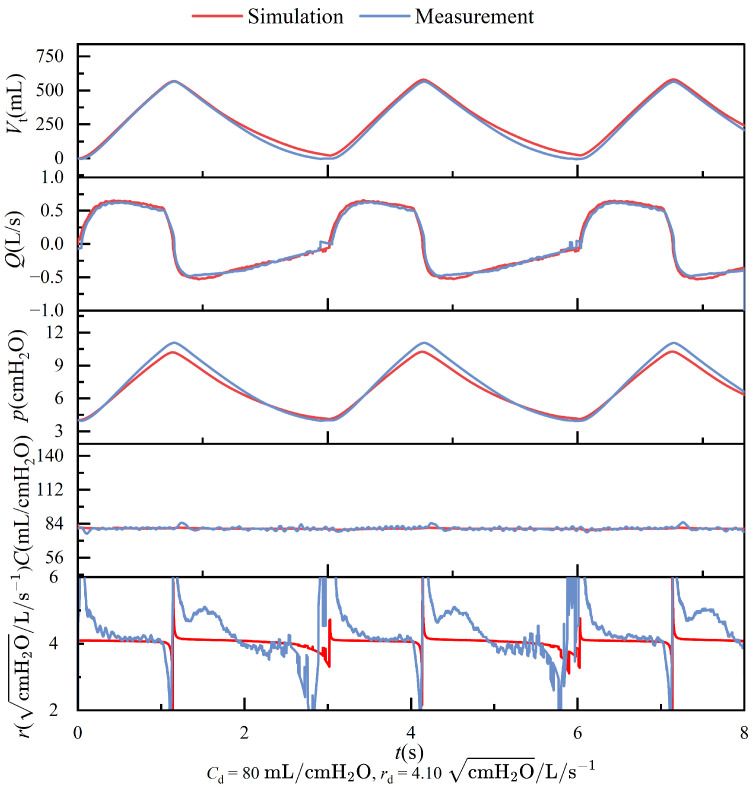
Simulation results in MATLAB/Simulink for breathing under a pressure supply mode with a frequency of 20 RR, compared with experimental observation data.

**Figure 11 bioengineering-12-00963-f011:**
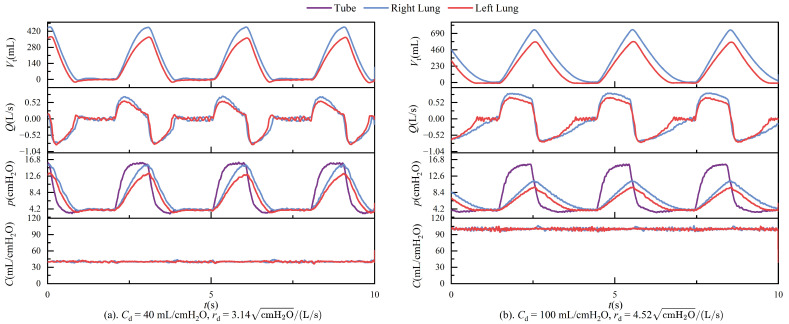
In passive mode, the left and right lungs’ tidal volume, flow, pressure, and compliance curves are under the same compliance pressure and resistance settings.

**Figure 12 bioengineering-12-00963-f012:**
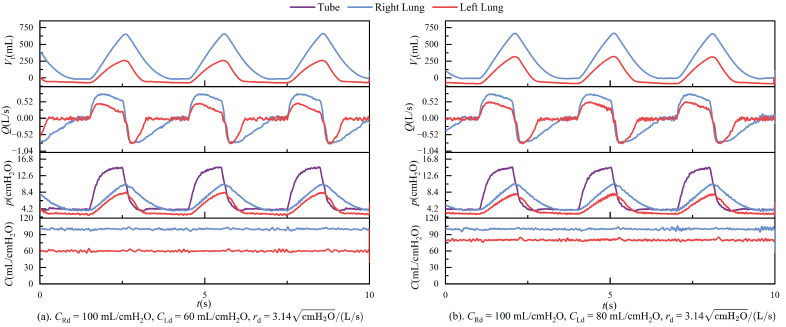
In passive mode, the left and right lungs’ tidal volume, flow, pressure, and compliance curves are under different compliance and resistance settings.

**Figure 13 bioengineering-12-00963-f013:**
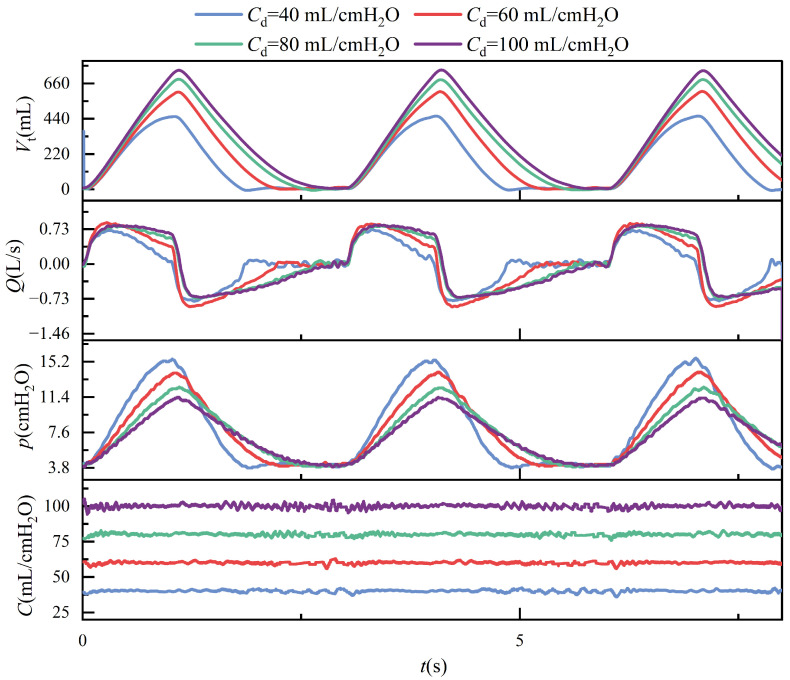
Time-dependent changes in tidal volume, flow, pressure, and compliance when $r_{d}$ = 3.14 $\;\sqrt{{\mathrm{cmH}}_{2}O}/\left(L/s\right)$ and $C_{d}$ = 40, 60, 80, and 100 mL/${\mathrm{cmH}}_{2}$O.

**Figure 14 bioengineering-12-00963-f014:**
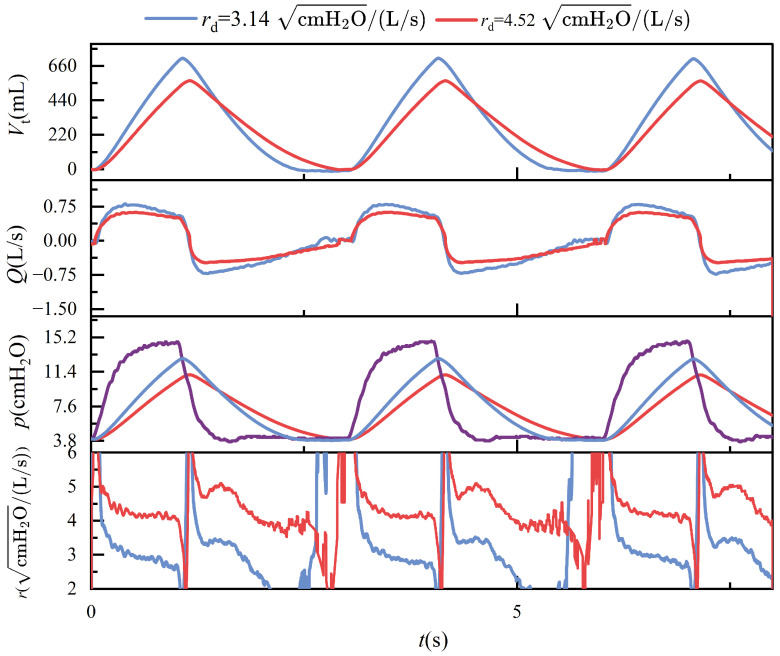
Time-dependent changes in tidal volume, flow, pressure, and compliance when $C_{d}$ = 80 mL/${\mathrm{cmH}}_{2}$O, $r_{d}$ = 3.14, and 4.54 $\;\sqrt{{\mathrm{cmH}}_{2}O}/\left(L/s\right)$.

**Figure 15 bioengineering-12-00963-f015:**
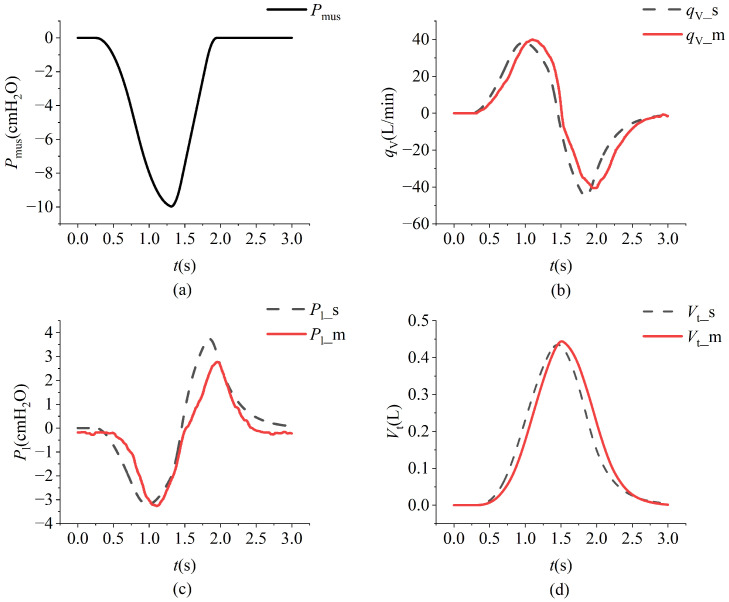
The variations in respiratory muscle force, intrapulmonary pressure, flow, and tidal volume over time under normal conditions. Espiratory muscle force (**a**), intrapulmonary pressure (**b**), flow (**c**), and tidal volume (**d**).

**Figure 16 bioengineering-12-00963-f016:**
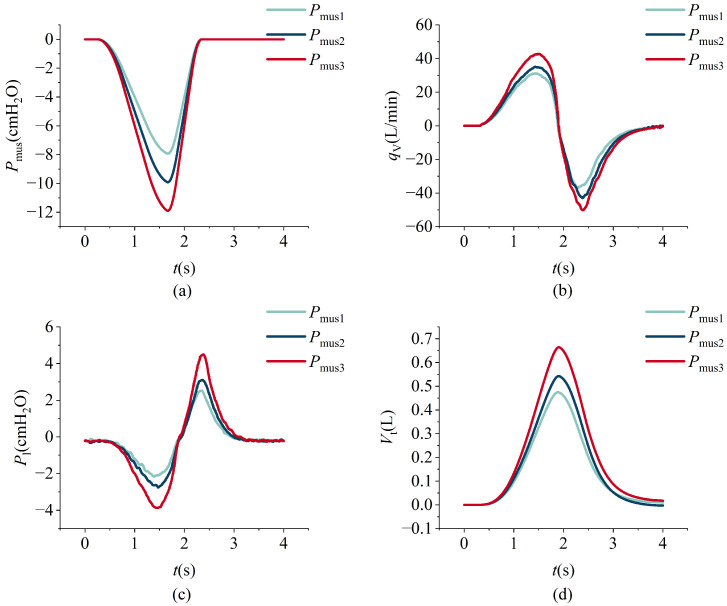
The variations in respiratory muscle force, intrapulmonary pressure, flow, and tidal volume over time under different respiratory muscle force conditions. Espiratory muscle force (**a**), intrapulmonary pressure (**b**), flow (**c**), and tidal volume (**d**).

**Figure 17 bioengineering-12-00963-f017:**
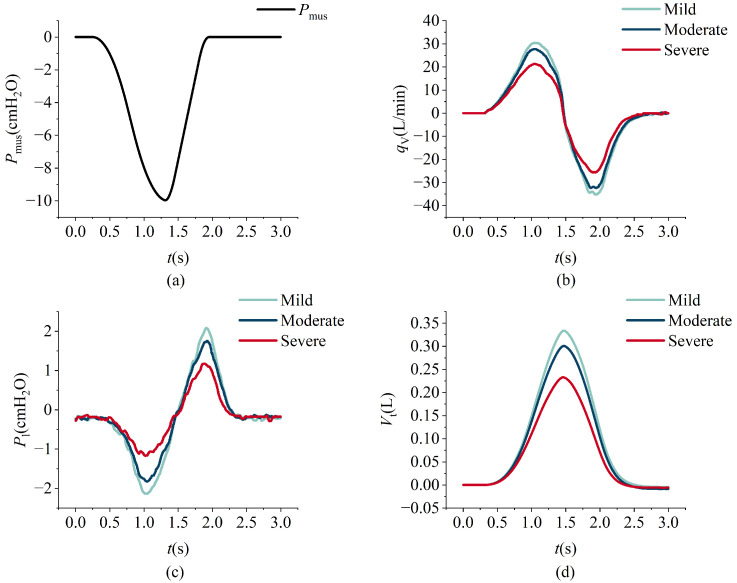
The variations in respiratory muscle force, intrapulmonary pressure, flow, and tidal volume over time under ARDS conditions. Espiratory muscle force (**a**), intrapulmonary pressure (**b**), flow (**c**), and tidal volume (**d**).

**Figure 18 bioengineering-12-00963-f018:**
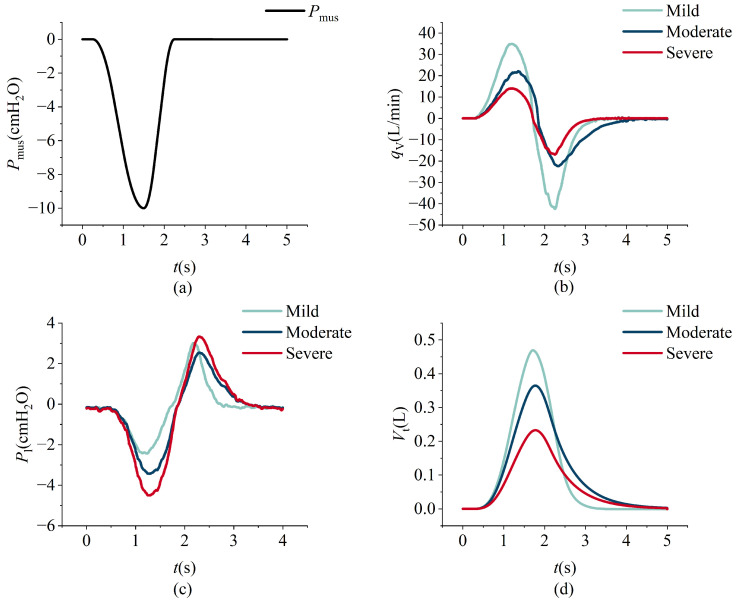
The variations in respiratory muscle force, intrapulmonary pressure, flow, and tidal volume over time under COPD conditions. Espiratory muscle force (**a**), intrapulmonary pressure (**b**), flow (**c**), and tidal volume (**d**).

**Figure 19 bioengineering-12-00963-f019:**
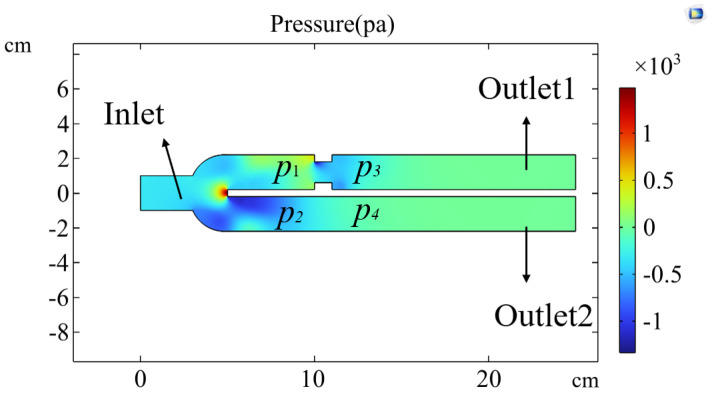
COMSOL Simulation Result. The isobar distribution of airflow passes through a three-way connector with unequal resistance on both sides at a constant speed.

**Figure 20 bioengineering-12-00963-f020:**
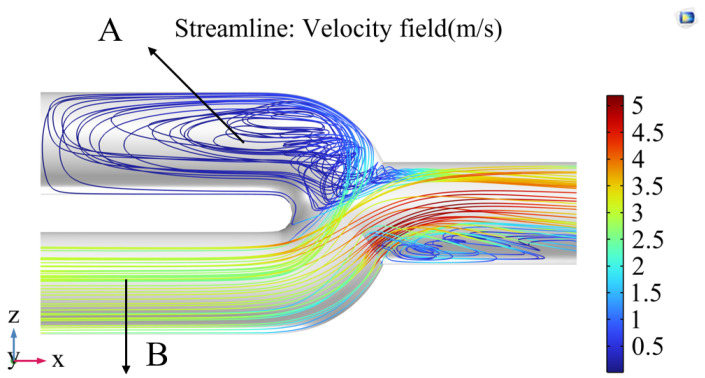
COMSOL Simulation Results of the Jet Phenomenon. The streamline of the velocity field when airflow enters the three-way connector at a constant speed from point B.

**Table 1 bioengineering-12-00963-t001:** The resistance coefficient corresponds to a lung resistance of 5 mbar/L/s in the SmartLung Adult lung simulator.

*Q* (L/s)	$\Delta p\;({\mathrm{cmH}}_{2}O)$	$r\;(\sqrt{{\mathrm{cmH}}_{2}O}/\left(L/s\right))$
0.33	1.13	3.22
0.50	2.41	3.10
0.66	4.28	3.13
0.83	6.61	3.10

**Table 2 bioengineering-12-00963-t002:** The resistance coefficient corresponds to a lung resistance of 20 mbar/L/s in the SmartLung Adult lung simulator.

*Q* (L/s)	$\Delta p({\mathrm{cmH}}_{2}O)$	$r(\sqrt{{\mathrm{cmH}}_{2}O}/\left(L/s\right))$
0.33	2.42	4.71
0.50	5.17	4.54
0.66	8.17	4.33
0.83	13.97	4.50

**Table 3 bioengineering-12-00963-t003:** Parameters of Sliding Mode Control.

Symbol	Physical Significance	Value
*k*	Sliding mode surface coefficient	10
$k_{1}$	Reaching law coefficient	25
$\beta_{1}$	Observer coefficient	100
$\beta_{2}$	Observer coefficient	300
$\alpha_{1}$	fal function constant	0.5
$\alpha_{2}$	fal function constant	0.25
δ	fal function constant	0.01

**Table 4 bioengineering-12-00963-t004:** Ventilator parameters. ($P_{peak}$ represents the peak inspiratory airway pressure, PEEP denotes the positive end-expiratory pressure, RR stands for respiratory rate, and TI refers to the inspiratory time).

Parameter	Value
$P_{peak}$	15 ${\mathrm{cmH}}_{2}$O
PEEP	4 ${\mathrm{cmH}}_{2}$O
RR	20 s
TI	1 s

**Table 5 bioengineering-12-00963-t005:** Measurement Compliance Over Three Cycles.

Target Compliance (mL/${\mathrm{cmH}}_{2}$O)	Average Value (mL/${\mathrm{cmH}}_{2}$O)	Measurement Range (mL/${\mathrm{cmH}}_{2}$O)	Error (%)
40	40.26	38.03–41.87	±4.93
60	60.12	57.12–62.66	±4.80
80	80.12	77.36–82.20	±3.3
100	100.08	96.59–102.44	±3.41

**Table 6 bioengineering-12-00963-t006:** Respiratory mechanics parameters under normal conditions.

$P_{\mathrm{mus}-\max}$	$C_{\mathrm{cw}}$	$C_{r}$	$C_{l}$	$R_{r}$	$R_{l}$	$R_{t}$	*r*	RR
**(${\mathrm{cmH}}_{2}$O)**	**(mL/${\mathrm{cmH}}_{2}$O)**	**(mL/${\mathrm{cmH}}_{2}$O)**	**(mL/${\mathrm{cmH}}_{2}$O)**	**(${\mathrm{cmH}}_{2}$O/(L/s))**	**(${\mathrm{cmH}}_{2}$O/(L/s))**	**(${\mathrm{cmH}}_{2}$O/(L/s))**	**($\sqrt{{\mathrm{cmH}}_{2}O}/\left(L/s\right)$)**	**(bpm)**
10	200	35	35	1	1	4	2.84	16

**Table 7 bioengineering-12-00963-t007:** Respiratory parameters under varying respiratory muscle effort.

Category	$P_{\mathrm{mus}-\max}$	$C_{\mathrm{cw}}$	$C_{r}$	$C_{l}$	$R_{r}$	$R_{l}$	$R_{t}$	*r*	RR
	**(${\mathrm{cmH}}_{2}$O)**	**(mL/${\mathrm{cmH}}_{2}$O)**	**(mL/${\mathrm{cmH}}_{2}$O)**	**(mL/${\mathrm{cmH}}_{2}$O)**	**(${\mathrm{cmH}}_{2}$O/(L/s))**	**(${\mathrm{cmH}}_{2}$O/(L/s))**	**(${\mathrm{cmH}}_{2}$O/(L/s))**	**($\sqrt{{\mathrm{cmH}}_{2}O}/\left(L/s\right)$)**	**(bpm)**
$P_{mus1}$	8	200	50	50	2	2	4	2.84	15
$P_{mus2}$	10	200	50	50	2	2	4	2.84	15
$P_{mus3}$	12	200	50	50	2	2	4	2.84	15

**Table 8 bioengineering-12-00963-t008:** Respiratory parameters in ARDS conditions.

Category	$P_{\mathrm{mus}-\max}$	$C_{\mathrm{cw}}$	$C_{r}$	$C_{l}$	$R_{r}$	$R_{l}$	$R_{t}$	*r*	RR
	**(${\mathrm{cmH}}_{2}$O)**	**(mL/${\mathrm{cmH}}_{2}$O)**	**(mL/${\mathrm{cmH}}_{2}$O)**	**(mL/${\mathrm{cmH}}_{2}$O)**	**(${\mathrm{cmH}}_{2}$O/(L/s))**	**(${\mathrm{cmH}}_{2}$O/(L/s))**	**(${\mathrm{cmH}}_{2}$O/(L/s))**	**($\sqrt{{\mathrm{cmH}}_{2}O}/\left(L/s\right)$)**	**(bpm)**
Mild	10	200	22.5	22.5	1	1	4	2.84	16
Moderate	10	200	20	20	2	2	4	2.84	16
Severe	10	200	15	15	4	4	4	2.84	16

**Table 9 bioengineering-12-00963-t009:** Respiratory parameters in COPD conditions.

Category	$P_{\mathrm{mus}-\max}$	$C_{\mathrm{cw}}$	$C_{r}$	$C_{l}$	$R_{r}$	$R_{l}$	$R_{t}$	*r*	RR
	**(${\mathrm{cmH}}_{2}$O)**	**(mL/${\mathrm{cmH}}_{2}$O)**	**(mL/${\mathrm{cmH}}_{2}$O)**	**(mL/${\mathrm{cmH}}_{2}$O)**	**(${\mathrm{cmH}}_{2}$O/(L/s))**	**(${\mathrm{cmH}}_{2}$O/(L/s))**	**(${\mathrm{cmH}}_{2}$O/(L/s))**	**($\sqrt{{\mathrm{cmH}}_{2}O}/\left(L/s\right)$)**	**(bpm)**
Mild	10	200	35	35	1	1	4	2.84	12
Moderate	10	200	35	35	2	2	10	4.42	12
Severe	10	200	20	20	8	8	20	5.93	12

**Table 10 bioengineering-12-00963-t010:** Measurement Compliance Over Three Cycles.

Category		Response Time (ms)
Normal		113
$P_{mus1}$		92
$P_{mus2}$		104
$P_{mus3}$		146
	Mild	106
ARDS	Moderate	97
	Severe	114
	Mild	180
COPD	Moderate	167
	Severe	174

## Data Availability

The real medical datasets are not available due to privacy restrictions.

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
