# Peer review of "Precision-Controlled Bionic Lung Simulator for Dynamic Respiration Simulation"

_bioengineering, 2025, doi:10.3390/bioengineering12090963_

Round 1
Reviewer 1 Report
Comments and Suggestions for Authors
The lung simulator manuscript provides a unique system that appears to overcome several design limitations of several lung simulators and dynamic respiration measurement systems on the market today. A few items the authors may wish to consider in review:
- The introduction does a commendable job providing an overview of the state-of the art of a wide variety of systems available to researchers. The authors may wish to consider a more pointed summary of the existing technology prior to the summary paragraph [Lines 85-90]. The section is so comprehensive, the reader can lose track of a summary of the most significant deficiencies prior to advocating their system as a solution.
- Figures 1-4 are a valuable addition to the manuscript, but they are so small and difficult to read, the authors may wish to exclude or format in a way that technical features are mor obvious.
- It seems a key feature of the proposed device is the dynamic control of the throttle opening, but this feature is not self-evident in Figure 6. It is unclear is the sliding surface is actually the same as the throttle opening.
- The descriptions of Sections 2.3.1 Passive Mode and 2.3.2 Active Mode are very completely described, and the detail builds confidence in reproducibility. To help the reader it may be beneficial to add a transition or way-finding summary at the start of Section 2.3 Control Methods to give the readers sense of what is most important in the two Sections that follow. It is easy to get lost in the detail and not recall clearly what is the core of the innovation being proposed.
- Line 260 refers to Figure 3 regarding the non-linear nature of the resistive element, but the Figure does not seem to inform as such.
- The MATLAB/Simulink platform is used to validate the hardware platform. It is unclear if this is a standard model (i.e.: https://www.mathworks.com/help/simscape/ug/medical-ventilator-with-lung-model.html) or what standards may have been used that other researchers might use as reference.
- Results appear to underscore the value of the current technique (Line 299 and Lines 314-316, Table 7). The Discussion section is also quite thorough, and the COMSOL analysis points out some design challenges, and it may help some readers to provide a smoother introduction to the sequence of results and relative importance. It is unclear whether COMSOL is critical to the design.
Author Response
The lung simulator manuscript provides a unique system that appears to overcome several design limitations of several lung simulators and dynamic respiration measurement systems on the market today. A few items the authors may wish to consider in review:
Reply:
Thank you very much for your valuable comments and suggestions. In this revision, we sharpen the motivation by adding a concise pre-closing synthesis in the Introduction that highlights the most consequential limitations of current simulators; reformat and enlarge Figures 1–4 to improve legibility of technical features; clarify in the text and in the caption of Figure 6 that the sliding surface s arises from LESO-based compliance tracking and is distinct from the micro-motor–actuated throttle opening, and redraw Figure 3 to label the gas-regulation unit and orifice throttle; add a brief way-finding transition at the start of Section 2.3 to orient readers before the passive/active subsections; explain, with reference to Figure 3, that the thin-walled orifice design induces turbulent, thus nonlinear, resistive behavior; and specify that our MATLAB/Simulink platform is an independently implemented model derived from eqs. (19), (20), and (25) with parameters from Table 3, rather than a Simscape example. We also smooth the Discussion to guide the sequence and relative importance of results. We also clarify that the COMSOL analysis is used to interpret left–right differences observed experimentally rather than to drive the core device design. The manuscript highlights all corresponding revisions with a yellow background; detailed responses follow.
1. The Introduction does a commendable job providing an overview of the state-of-the-art of a wide variety of systems available to researchers. The authors may wish to consider a more pointed summary of the existing technology prior to the summary paragraph . The section is so comprehensive, the reader can lose track of a summary of the most significant deficiencies prior to advocating their system as a solution.
Reply:
We sincerely thank the reviewer for this valuable comment. In response, we have added a concise, focused summary of the most significant deficiencies in current lung simulators immediately before the concluding paragraph of the Introduction, thereby improving flow and making the research gap explicit. The added text has been highlighted in the manuscript.
2. Figures 1-4 are a valuable addition to the manuscript, but they are so small and difficult to read, the authors may wish to exclude or format in a way that technical features are mor obvious.
Reply:
We sincerely thank the reviewer for this valuable comment. We have reformatted and enlarged Figures 1–4 to enhance readability and to emphasize key technical features.
3. It seems a key feature of the proposed device is the dynamic control of the throttle opening, but this feature is not self-evident in Figure 6. It is unclear is the sliding surface is actually the same as the throttle opening.
Reply:
We sincerely thank the reviewer for this valuable comment. As clarified in the revised text and the caption of Figure 6, the sliding surface s is defined from the compliance tracking error (LESO state). Figure 6 depicts the LESO–SMC compliance control block diagram. A micro-motor in the gas regulation unit actuates the throttle opening. However, it is not the same variable as s, it is coupled to the throttling element at the system level and jointly shapes the pressure–flow–volume dynamics. To further avoid ambiguity, we have redrawn Figure 3 and explicitly annotated the micro-motor and throttling component. The associated revisions have been highlighted in the manuscript.
4. The descriptions of Sections 2.3.1 Passive Mode and 2.3.2 Active Mode are very completely described, and the detail builds confidence in reproducibility. To help the reader it may be beneficial to add a transition or way-finding summary at the start of Section 2.3 Control Methods to give the readers sense of what is most important in the two Sections that follow. It is easy to get lost in the detail and not recall clearly what is the core of the innovation being proposed.
Reply:
We sincerely thank the reviewer for this valuable comment. We have inserted a concise transition summary at the beginning of Section 2.3 to guide the reader and emphasize the core innovations before delving into the passive and active control modes. The added summary has been highlighted in the manuscript.
5. Line 260 refers to Figure 3 regarding the nonlinear nature of the resistive element, but the Figure does not seem to inform as such.
Reply:
We sincerely thank the reviewer for this insightful suggestion. As illustrated in Figure 3, the resistance regulation device adopts a thin-walled orifice-type design, which induces turbulence as airflow passes through and causes the entire throttling element to exhibit pronounced nonlinear characteristics. We have supplemented the manuscript with this clarification, and the revised content has been highlighted accordingly.
6. The MATLAB/Simulink platform is used to validate the hardware platform. It is unclear if this is a standard model (i.e.: https://www.mathworks.com/help/simscape/ug/medical-ventilator-with-lung-model.html) or what standards may have been used that other researchers might use as reference.
Reply:
We sincerely thank the reviewer for raising this critical point. Our MATLAB/Simulink simulation platform was not built upon existing Simscape medical ventilator examples. Instead, the simulation model was independently developed based on equation (25), with the compliance control implemented according to the discretized formulas (19) and (20), and parameters set in reference to Table 3. The airway resistance regulation module was realized according to equation (3). We have added a clarification in the manuscript to state the origin and implementation details of the model explicitly. The revised content has been highlighted in the updated manuscript.
7. Results appear to underscore the value of the current technique (Line 299 and Lines 314-316, Table 7). The Discussion section is also quite thorough, and the COMSOL analysis points out some design challenges, and it may help some readers to provide a smoother introduction to the sequence of results and relative importance. It is unclear whether COMSOL is critical to the design.
Reply:
We sincerely thank the reviewer for this valuable comment. The COMSOL section was primarily included to explain the differences observed between the left and right lungs during the experiments, which are attributable to structural factors. In response to the suggestion, we have reorganized the presentation of the results in the Discussion section to improve clarity.
Reviewer 2 Report
Comments and Suggestions for Authors
The paper describes a dual-mode, precision-controlled bionic lung simulator for active and passive respiratory emulation. The work presents a complex concept because it integrates a sliding mode controller with an extended state linear observer to allow for fine-tuning of compliance and resistance. However, the methodological descriptions lack the necessary details for replicability, particularly regarding the tuning of control parameters and the specifications of mechanical components. The figures lack proper annotation. The method for estimating airway resistance is known to be unstable at low flows, but no mitigation strategy is attempted. The authors must integrate and improve the study according to the following observations.
1. How did the authors calibrate and validate the spatial distribution of pressure within the lung chambers during dynamic breathing cycles? Without spatial verification, localised variations can affect the accuracy of compliance. The authors should integrate the introduction, after discussing the limitations of existing simulators in replicating realistic respiratory dynamics, with recent studies on smart electronic device-based monitoring of SAR and temperature variations in human tissue interaction indoors. These studies provide methods for spatially resolving heat/SAR models in tissue-equivalent models, which could be adapted to monitor non-uniform mechanical loads in the lung balloon.
2. The yield and resistance elements are treated as concentrated parameters, but potential internal material degradation during repeated cycles is not considered. Could the authors implement non-destructive testing with eddy currents for monitoring the structural health of the airbag and its resistance components? Studies on soft computing techniques and eddy currents to estimate and classify delaminations in the CFRP plates of biomedical devices could detect microstructural changes in the polymer or composite components of the simulator, improving long-term reliability assessment.
3. Given the extensive experimental dataset, did the authors consider applying deep learning time series models to classify ventilation modes and automatically detect anomalies? I'm not asking the authors to implement a model from scratch, but only to integrate the Discussion section and the bibliography with the study doi: 10.1109/TIM.2025.3573363 because LSTM-U-Net architectures are suitable for spatiotemporal biomedical signals; here, they could segment the respiratory phases and detect patterns of patient-ventilator asynchrony.
4. The simulator focuses on respiratory dynamics, but musculoskeletal interactions are minimally modelled. Could the authors integrate finite element modelling of the chest wall for greater realism? Recent studies on FEM modelling and an enhanced monitoring system for rehabilitation have demonstrated how FEM can model complex biomechanical deformations.
5. In the proposed control system, signal acquisition is fundamental for stability. Did the authors evaluate high-sensitivity optoelectronic sensors or avalanche diodes to improve flow/pressure measurement under weak signal conditions?
6. Modelling airway resistance at low flows remains unstable. Could the authors incorporate finite element-based turbulence modelling to refine the drag coefficient calculation?
7. Did the authors consider integrating multimodal sensors to acquire tissue-ventilator interactions in addition to pressure and flow, such as impedance tomography for lung volume distribution?
8. Could muscle strength profiles in active mode be cross-validated with EMG-guided simulations to increase physiological accuracy?
9. The study would benefit from home care systems or portable ventilation. Did the authors consider including comparative data? The development of a non-invasive ventilator for home care and a patient monitoring system provides benchmarks for the design and validation of non-invasive ventilation systems that could serve as external comparators.
10. Could the mechanical fatigue of the airbag and resistance device under prolonged cycles be analysed using fuzzy logic-based stress assessment? Recent studies have delved into the intuitionistic fuzzy divergence for assessing the mechanical stress state of steel plates subjected to biaxial loads. The fuzzy divergence approach can handle uncertain stress states and could be adapted for elastomer fatigue prediction.
11. Could the authors delve deeper into the study of the use of integrated electronics and artificial intelligence for real-time maintenance alerts within the simulator system?
12. Did the authors consider ultrasonic energy to model the humidified airflow and condensation in the simulator? The hybrid analytical-numerical approach could model the thermal and fluid dynamic effects relevant to the simulation of humidified breathing.
Author Response
The paper describes a dual-mode, precision-controlled bionic lung simulator for active and passive respiratory emulation. The work presents a complex concept because it integrates a sliding mode controller with an extended state linear observer to allow for fine-tuning of compliance and resistance. However, the methodological descriptions lack the necessary details for replicability, particularly regarding the tuning of control parameters and the specifications of mechanical components. The figures lack proper annotation. The method for estimating airway resistance is known to be unstable at low flows, but no mitigation strategy is attempted. The authors must integrate and improve the study according to the following observations.
Reply:
Thank you very much for your valuable comments and suggestions. This work is positioned as a bench-top, dual-chamber simulator with active/passive modes and closed-loop control of compliance and resistance using LESO-SMC, and the revision keeps the focus on global respiratory mechanics and controller stability while responding to the reviewer's concerns in a consolidated way. We clarify the study's scope and limitations up front, strengthen the Discussion to outline concrete, staged extensions—spatial verification via CFD cross-checks and optional EIT/multi-point sensing, material health monitoring paths suited to future conductive composites, offline/weakly coupled FEM chest-wall modelling, optional EMG-guided validation, predictive maintenance based on telemetry, and ultrasonic/hybrid modelling for humidified breathing—and we document a practical mitigation strategy for low-flow resistance estimation while reserving CFD/FEM turbulence analyses for offline calibration rather than real-time control. We also add the requested reference on LSTM–U-Net and note our plan for a standardized data/label interface to facilitate downstream analytics. We have revised the paper accordingly; the modified sections are highlighted with yellow. Please find our responses to your comments below.
1. How did the authors calibrate and validate the spatial distribution of pressure within the lung chambers during dynamic breathing cycles? Without spatial verification, localised variations can affect the accuracy of compliance. The authors should integrate the Introduction, after discussing the limitations of existing simulators in replicating realistic respiratory dynamics, with recent studies on smart electronic device-based monitoring of SAR and temperature variations in human tissue interaction indoors. These studies provide methods for spatially resolving heat/SAR models in tissue-equivalent models, which could be adapted to monitor non-uniform mechanical loads in the lung balloon.
Reply:
We appreciate the suggestion and clarify that the present work targets global mechanics and closed-loop stability; spatial pressure mapping is beyond scope but now explicitly acknowledged as a limitation, and we outline two future extensions: (i) CFD-based cross-checks under measured boundary conditions and (ii) integration of multi-point pressure sensors and/or EIT via reserved hardware/timing interfaces. Drawing on device-based spatial monitoring paradigms (e.g., SAR/temperature fields), these steps will enable localized assessment of non-uniform mechanical loads. The clarification and planned directions have been highlighted in the manuscript.
2. The yield and resistance elements are treated as concentrated parameters, but potential internal material degradation during repeated cycles is not considered. Could the authors implement non-destructive testing with eddy currents for monitoring the structural health of the airbag and its resistance components? Studies on soft computing techniques and eddy currents to estimate and classify delaminations in the CFRP plates of biomedical devices could detect microstructural changes in the polymer or composite components of the simulator, improving long-term reliability assessment.
Reply:
We sincerely thank the reviewer for this valuable comment. We agree that structural health monitoring is essential; because our airbag and resistance elements are polymer/elastomer, conventional eddy-current NDT (optimized for conductive media) is not directly applicable at present, but if future designs adopt conductive composites we will evaluate eddy-current and ultrasonic NDT for in-situ health tracking; these material constraints and future pathways are now stated in the Discussion and noted in the Appendix specifications, with changes highlighted.
3. Given the extensive experimental dataset, did the authors consider applying deep learning time series models to classify ventilation modes and automatically detect anomalies? I'm not asking the authors to implement a model from scratch, but only to integrate the Discussion section and the bibliography with the study doi: 10.1109/TIM.2025.3573363 because LSTM-U-Net architectures are suitable for spatiotemporal biomedical signals; here, they could segment the respiratory phases and detect patterns of patient-ventilator asynchrony.
Reply:
We sincerely thank the reviewer for this valuable comment. We agree that LSTM–U-Net-style architectures are well-suited to segment respiratory phases and detect patient–ventilator asynchrony. We have: (i) stated in the Discussion that we will establish a standardized data/label interface and logging protocol to facilitate such downstream applications, and (ii) added the suggested reference (doi:10.1109/TIM.2025.3573363) to the bibliography. These updates have been highlighted in the manuscript.
4. The simulator focuses on respiratory dynamics, but musculoskeletal interactions are minimally modelled. Could the authors integrate finite element modelling of the chest wall for greater realism? Recent studies on FEM modelling and an enhanced monitoring system for rehabilitation have demonstrated how FEM can model complex biomechanical deformations.
Reply:
We sincerely thank the reviewer for this valuable comment. The prototype uses a lumped-parameter chest-wall model to preserve interpretability and real-time control. Full FEM coupling would compromise responsiveness and computational simplicity at this stage. We recognize FEM's potential for posture/pathology-specific studies and plan to explore an offline/weakly coupled interface via boundary-condition exchange in future work. This perspective has been highlighted in the manuscript.
5. In the proposed control system, signal acquisition is fundamental for stability. Did the authors evaluate high-sensitivity optoelectronic sensors or avalanche diodes to improve flow/pressure measurement under weak signal conditions?
Reply:
We sincerely thank the reviewer for this valuable comment. Weak signals at low flow are stabilized via modest hardware averaging, rate limiting, and lightweight digital filtering, which suffices for closed-loop operation without altering dynamics. While higher-sensitivity sensors could benefit extreme conditions, they are not required for the present results. The architecture remains compatible with incremental sensor upgrades.
6. Modelling airway resistance at low flows remains unstable. Could the authors incorporate finite element-based turbulence modelling to refine the drag coefficient calculation?
Reply:
We sincerely thank the reviewer for this valuable comment. The instability at low flow primarily arises from the denominator term involving flow rate and noise amplification in transitional regimes. We acknowledge the value of finite element–based turbulence modelling for improving drag coefficient estimation; however, it is presently regarded as a tool for offline calibration rather than online computation. This direction will be further considered in future work to enhance the physical fidelity of the model.
7. Did the authors consider integrating multimodal sensors to acquire tissue-ventilator interactions in addition to pressure and flow, such as impedance tomography for lung volume distribution?
Reply:
We sincerely thank the reviewer for this valuable comment. While the present study relies on single-point pressure and flow sensing for global compliance estimation and closed-loop stability, we fully acknowledge the potential of multimodal sensors for capturing tissue–ventilator interactions. In our system design, hardware and timing interfaces have been reserved for the future integration of electrical impedance tomography (EIT) and multi-point pressure sensors, which could enable lung volume distribution monitoring and spatial consistency verification of compliance. These extensions are planned as future developments to enhance physiological fidelity further.
8. Could muscle strength profiles in active mode be cross-validated with EMG-guided simulations to increase physiological accuracy?
Reply:
We sincerely thank the reviewer for this valuable comment. At this stage, constraints in hardware interfaces, human-subject approvals, and data availability prevent us from incorporating EMG-guided cross-validation into the present study. Our current focus is on device and control performance, where the active mode allows externally defined Pmus trajectories. These provide a parameterized and repeatable representation of patient effort, consistent with our goals of real-time operation, identifiability, and closed-loop stability. To ensure extensibility without altering the present scope, we have reserved interfaces for synchronized EMG and other biosignals, and we acknowledge this as a limitation to be addressed in future work.
9. The study would benefit from home care systems or portable ventilation. Did the authors consider including comparative data? The development of a non-invasive ventilator for home care and a patient monitoring system provides benchmarks for the design and validation of non-invasive ventilation systems that could serve as external comparators.
Reply:
We sincerely thank the reviewer for this valuable comment. The present study is primarily positioned as a bench-top evaluation and research–teaching platform. Comparative data with home care systems or portable ventilators were not included. However, we recognize their importance as external benchmarks, and in future work, we plan to explore integration with non-invasive ventilation systems and home-care devices to support design and validation from a broader translational perspective. The related Discussion has been highlighted in the revised manuscript.
10. Could the mechanical fatigue of the airbag and resistance device under prolonged cycles be analysed using fuzzy logic-based stress assessment? Recent studies have delved into the intuitionistic fuzzy divergence for assessing the mechanical stress state of steel plates subjected to biaxial loads. The fuzzy divergence approach can handle uncertain stress states and could be adapted for elastomer fatigue prediction.
Reply:
We sincerely thank the reviewer for this valuable comment. Currently, the lack of long-cycle stress–damage data prevents fuzzy logic fatigue assessment. As an engineering substitute, we propose periodic cyclic fatigue tests combined with compliance/leakage baseline re-measurement. In the longer term, intuitionistic fuzzy divergence methods may be applied for elastomer fatigue prediction with extended datasets. The related Discussion has been highlighted in the revised manuscript.
11. Could the authors delve deeper into the study of the use of integrated electronics and artificial intelligence for real-time maintenance alerts within the simulator system?
Reply:
We sincerely thank the reviewer for this valuable comment. The system already incorporates remote telemetry and logging capabilities. As a prototype solution, we have proposed threshold-based and statistical process control (SPC) alerting within the architecture. Looking ahead, with sufficient data accumulation, predictive maintenance strategies based on drift detection and anomaly distribution analysis will be introduced. These enhancements are planned as future work and do not affect the experimental validation presented in this study. The related Discussion has been highlighted in the revised manuscript.
12. Did the authors consider ultrasonic energy to model the humidified airflow and condensation in the simulator? The hybrid analytical-numerical approach could model the thermal and fluid dynamic effects relevant to the simulation of humidified breathing.
Reply:
We thank the reviewer for this interesting suggestion. The present study focuses on mechanical compliance and resistance control, and humidification–condensation effects were not incorporated. We acknowledge, however, that ultrasonic energy and hybrid analytical–numerical modelling provide promising tools to capture thermal and fluid dynamic phenomena of humidified breathing. These approaches will be considered in future simulator extensions to broaden physiological fidelity.

Round 2
Reviewer 2 Report
Comments and Suggestions for Authors
I thank the authors for their replies to the comments. I have no further comments to make.